# DNA Methylation Patterns in Relation to Acute Severity and Duration of Anxiety and Depression

Eva Vidovič [1], Sebastian Pelikan [1], Marija Atanasova [2], Katarina Kouter [3,4], Indre Pileckyte [5], Aleš Oblak [1], Brigita Novak Šarotar [1,6], Alja Videtič Paska [3,*] and Jurij Bon [1,6]

[1] University Psychiatric Clinic Ljubljana, 1260 Ljubljana, Slovenia; jurij.bon@mf.uni-lj.si (J.B.)
[2] Faculty of Chemistry and Chemical Technology, University of Ljubljana, 1000 Ljubljana, Slovenia
[3] Institute for Biochemistry and Molecular Genetics, Faculty of Medicine, University of Ljubljana, 1000 Ljubljana, Slovenia
[4] Institute of Microbiology and Immunology, Faculty of Medicine, University of Ljubljana, 1000 Ljubljana, Slovenia
[5] Center for Brain and Cognition, Pompeu Fabra University, 08018 Barcelona, Spain
[6] Department of Psychiatry, Faculty of Medicine, University of Ljubljana, 1000 Ljubljana, Slovenia
* Correspondence: alja.videtic@mf.uni-lj.si

**Abstract:** Depression and anxiety are common mental disorders that often occur together. Stress is an important risk factor for both disorders, affecting pathophysiological processes through epigenetic changes that mediate gene–environment interactions. In this study, we explored two proposed models about the dynamic nature of DNA methylation in anxiety and depression: a stable change, in which DNA methylation accumulates over time as a function of the duration of clinical symptoms of anxiety and depression, or a flexible change, in which DNA methylation correlates with the acute severity of clinical symptoms. Symptom severity was assessed using clinical questionnaires for anxiety and depression (BDI-II, IDS-C, and HAM-A), and the current episode and the total lifetime symptom duration was obtained from patients' medical records. Peripheral blood DNA methylation levels were determined for the *BDNF*, *COMT*, and *SLC6A4* genes. We found a significant negative correlation between *COMT_1* amplicon methylation and acute symptom scores, with BDI-II ($R(22) = 0.190$, $p = 0.033$), IDS-C ($R(22) = 0.199$, $p = 0.029$), and HAM-A ($R(22) = 0.231$, $p = 0.018$) all showing a similar degree of correlation. Our results suggest that DNA methylation follows flexible dynamics, with methylation levels closely associated with acute clinical presentation rather than with the duration of anxiety and depression. These results provide important insights into the dynamic nature of DNA methylation in anxiety and affective disorders and contribute to our understanding of the complex interplay between stress, epigenetics, and individual phenotype.

**Keywords:** anxiety; depression; epigenetics; DNA methylation; candidate genes; *BDNF*; *COMT*; *SLC6A4*





## 1. Introduction

Mental health is critical for overall well-being as it affects all areas of life, including work, relationships, social functioning, and physical health. Mental disorders often co-occur with other illnesses, making them an important comorbid factor that can worsen health outcomes [1]. Today, mental disorders are the second leading cause of the number of years lived with disability, with depressive and anxiety disorders being the most common across all age groups [2].

Depression and anxiety are thought to have multiple causes. Both occur clustered in the same families, although no gene has yet been shown to cause either disorder [3]. The latter has led researchers to focus on the external and internal environments as possible modulators of signaling pathways involved in affective disorders [4,5]. Research has shown that excessive stress exposure, failure of stress reduction mechanisms, chronic stress exposure, or exposure during sensitive periods are associated with compensatory overload

and maladaptive responses that lead to many neuropsychiatric disorders [6–8]. Stress is therefore an important risk factor influencing the pathophysiological processes of mental disorders [9], especially depressive and anxiety disorders.

There is increasing evidence that a common mechanism underlies these depressive and anxiety disorders, which calls into question the need for a strict distinction between these clinical categories [10,11]. Both disorders involve dysregulation of neurotransmitter systems, such as serotonin and norepinephrine, which are targeted using standard antidepressants [12]. In addition, many people with depression also suffer from anxiety symptoms and vice versa, indicating a significant overlap in symptomatology [13]. This shared pathophysiology suggests that treatments targeting common mechanisms may be effective in both disorders.

Once stress has been recognized by the individual, activation of the hypothalamic–pituitary–adrenocortical axis leads to the release of glucocorticoids, resulting in various changes, including epigenetic modifications of genes and chromatin remodeling [14,15]. Epigenetic mechanisms that modulate gene expression in response to environmental stimuli are one way environmental factors shape an organism's phenotype [16]. Neuronal differentiation has been shown to be closely linked to epigenetic mechanisms, such as DNA methylation, histone modifications, chromatin remodeling, and non-coding RNA [17]. Alterations in these mechanisms have been linked with the onset of neurodegenerative changes [18,19] and psychiatric disorders, including depression and anxiety [20].

A prerequisite for epigenetic modifications is the presence of the so-called epigenome editing tools, which allow for DNA methylation induction as well as deletion. This is performed using enzymes, such as DNA methyltransferases (DNMTs) and ten-eleven translocation (TET) methylcytosine dioxygenases. De novo methylation is catalyzed by DNA methyltransferase [21] and maintained by maintenance methyltransferase (DNMT1), which is involved in the replicative transfer of epigenetic information via mitosis [22]. A process that is diametrically opposed to DNA methylation is demethylation, which can occur passively, through the loss of a methyl group during mitosis, or actively, through TET enzymes [21]. For DNA methylation to be inherently dynamic, epigenome editing tools would also theoretically need to respond to environmental factors and/or stress. It has been well established that the regulation of epigenome editing tools responds to various environmental factors as well as stress, resulting in either upregulation or downregulation [23–25].

In an effort to understand the biological basis of mental disorders mediated by stress, researchers are focusing on DNA methylation studies of candidate genes, particularly those involved in monoamine transmission and neuroplasticity. The monoamine hypothesis states that dysregulations of monoamine neurotransmitters, such as serotonin, dopamine, and norepinephrine, underlie the development of anxiety and depression [1,26]. Strong candidate genes involved in the regulation of monoamines include the *SLC6A4* gene, which encodes the serotonin transporter, and the *COMT* gene, which encodes the enzyme catechol-O-methyltransferase, responsible for the degradation of catecholamines [3]. Similarly, the neuroplasticity theory suggests that impaired neuroplasticity, particularly decreased neurotrophic factor (BDNF) expression, may also contribute to the pathophysiology of these disorders [27,28]. Several meta-analyses examining gene polymorphisms [3,29–31] and methylation studies [32–34] have confirmed the important role of these candidate genes in the pathophysiology of stress-mediated disorders, such as anxiety and depression.

In addition, the timing and duration of stress exposure have been shown to play a critical role in the development of stress-related disorders. Several studies have linked stress exposure during specific sensitive periods to psychopathological symptoms later in life [35], with specific methylation patterns observed in some candidate genes [36–40]. There is a large body of research addressing the effects of chronic stress exposure, usually of low intensity, on the development of stress-related disorders [32,39]. This prompted Zannas and Chrousos [15] to propose two different models for the dynamics of stress-induced epigenetic changes. Their first model assumed the existence of sensitive periods during

life stages characterized by increased epigenetic plasticity, such as in utero, childhood, and adolescence, when protective mechanisms may be impaired [41,42]. During these periods of increased plasticity, the response to acute stressors may lead to more significant epigenetic changes. On the other hand, their second model assumed a cumulative effect of stress, in which repeated exposures to stressors gradually accumulate and affect the epigenome over time. Consequently, epigenetic changes vary as a function of the duration of stress exposure [15]. Both models emphasized the critical importance of considering the timing and intensity as well as the duration of stress exposure when studying the effects of stress on human health.

In this study, we investigated DNA methylation dynamics in a clinical context. The *BDNF*, *COMT*, and *SLC6A4* genes were selected as they have been previously implicated in signaling pathways that are potentially disrupted in depression and can therefore be considered candidate genes for depression. In addition, these three genes have the potential to interact with each other [34,43–45]. We wanted to determine whether epigenetic changes, such as DNA methylation patterns of the candidate genes (*BDNF*, *COMT*, and *SLC6A4*) in peripheral blood, correlate with the duration and acute severity of anxiety and depression symptoms measured with clinical scales (BDI-II, HAM-A, and IDS-C). This would inform us whether DNA methylation dynamics are different in chronic depression than in acute anxiety and depression. Based on the results of previous studies, we assumed that DNA methylation would conform to one of our proposed models. Under the flexible model, we expected that DNA methylation of our candidate genes would vary depending on the acute severity of anxiety and depression symptoms measured using clinical scales. This would imply that stress-induced changes during periods of increased symptom severity could lead to observable changes in the methylation levels. In contrast, in the case of the stable model, we assumed that DNA methylation would change as a function of the duration of stress exposure and that cumulative effects would lead to increasing changes in methylation patterns over time. As mentioned earlier, depression and anxiety symptoms often co-occur. Anxiety has previously been observed to impair responses to treatment for depression, leading to poorer outcomes, delayed or incomplete remission, and increased risk of relapse [46–48]. Given these findings, and the possibility that clinical symptomatology changes over the course of the disease, this study also examined whether individual components of the clinical picture (anxiety or depression) have a greater impact on the methylation levels.

## 2. Materials and Methods

### 2.1. Participants

The study included 25 Caucasian patients (16 women), aged 28 to 67 years (mean age 51 years), admitted to the University Psychiatric Clinic Ljubljana due to a clinically manifested depressive disorder or anxiety disorder. Diagnoses were confirmed using ICD-10 criteria by trained psychiatrists. Inclusion criteria were a diagnosis of depressive disorder (F32 and F33), anxiety, or stress disorder (F40, F41, and F43), age over 18 years, and written consent to participate in the study. Exclusion criteria were bipolar disorder, psychotic disorder, substance use disorder, or significant systemic or neurological disease that might affect brain function. All subjects were taking medication at the time of this study. One participant was excluded from regression analysis due to missing methylation data on the COMT_1 amplicon.

The study was approved by the Medical Ethics Committee and all participants signed an informed consent form (see the Institutional Review Board Statement).

### 2.2. Study Design

Clinical questionnaires included the Beck Depression Inventory (BDI-II [49]) to assess the patient's subjective experience of depressive symptoms; meanwhile, the Inventory of Depressive Symptomatology, Clinician version (IDS-C [50]) was used for an objective assessment of depressive symptoms. In addition, the Hamilton Anxiety Rating Scale (HAM-

A [51]) was used for objective assessment of anxiety symptoms, including psychological and somatic symptoms, and patient behavior during the psychiatric interview. For all clinical questionnaires, higher scores corresponded to a higher clinical severity of symptoms. The duration of the disorder was obtained from medical records and divided into the duration of the current clinical episode and the lifetime duration, excluding periods of symptom remission. Venous blood samples were obtained from all participants at the time of clinical assessment for analysis of methylation patterns. The clinical details of the participants can be found in Table S1 (Supplementary Materials).

*2.3. DNA Methylation Analysis*

2.3.1. Selection of Candidate Genes and Target Sequences

DNA methylation analysis was performed for the *BDNF*, *COMT*, and *SLC6A4* genes, which were selected for their associations with stress, anxiety, and depressive disorders, focusing on monoamine systems and neuroplasticity. The densest DNA methylation is usually found in the so-called CpG islands (CGIs), which are located in the promoter region and/or in the first intron. To locate these regions and map the target sequences in each gene of interest, we used the UCSC Genome Browser, Human Genome Build 19 (hg19) [52]. The target sequences in the candidate genes were located in the CGIs, with additional 500 base-pair flanking regions present upstream and downstream of the CGIs. When these genes lacked the specific CGIs, the transcription start site containing sequences with a potential regulatory function was used as the target sequence.

These genes and their coordinates in the human genome are given in Table 1.

**Table 1.** Amplicon positions of candidate genes in the human genome, according to Human Genome Build 19 (hg19).

| Amplicon | Position (Human Genome Build 19) and Length of the Target Sequence | Number of CpG Islands | Functional Significance |
|---|---|---|---|
| *BDNF_1* | chr11:27744260–27744605 (−), 346 | 22 | |
| *BDNF_2* | chr11:27743702–27743960 (−), 259 | 10 | |
| *BDNF_3* | chr11:27743454–27743762 (−), 309 | 20 | The brain-derived neurotrophic factor |
| *BDNF_4* | chr11:27741988–27742250 (−), 263 | 13 | regulates growth, differentiation, |
| *BDNF_5* | chr11:27740916–27741131 (−), 216 | 16 | maintenance, death/survival, and |
| *BDNF_6* | chr11:27740607–27740901 (−), 295 | 30 | plasticity of neurons [53] |
| *BDNF_7* | chr11:27721638–27721854 (−), 217 | 19 | |
| *BDNF_8* | chr11:27722466–27722696 (−), 231 | 13 | |
| *BDNF_9* | chr11:27722209–27722487 (−), 279 | 23 | |
| *COMT_1* | chr22:19951071–19951343, 273 | 14 | The catechol-O-methyl transferase |
| *COMT_2* | chr22:19929042–19929349, 308 | 36 | (COMT) determines prefrontal |
| *COMT_3* | chr22:19950002–19950320, 319 | 13 | dopaminergic availability [54] |
| *SLC6A4_1* | chr17:28562753–28563050 (−), 298 | 29 | The serotonin transporter is associated |
| *SLC6A4_2* | chr17:28563277–28563552 (−), 276 | 7 | with depression treatment outcomes [55] |

2.3.2. DNA Isolation and Bisulfite Conversion

Venous blood samples were coded to ensure data confidentiality and stored at −70 °C until DNA isolation. DNA isolation from the blood samples was performed using the DNA Mini Kit (Qiagen, Venlo, The Netherlands), according to the manufacturer's instructions. Gene methylation analysis was facilitated through DNA-to-bisulfite conversion, in which DNA is treated with sodium bisulfite. This causes the deamination of cytosine (C), converting cytosine into uracil (U). The presence of methylation makes the conversion into U impossible, leaving a C behind. A total of 1000 ng of DNA was converted into bisulfite using the EpiTect Fast Bisulfite Kit (Qiagen, Venlo, The Netherlands) and eluted in 50 μL of elution buffer, yielding purified DNA at a final concentration of 20 ng/μL.

### 2.3.3. Primer Design

After bisulfite conversion, polymerase chain reaction (PCR)-based methods were used to study DNA methylation [56,57]. Primers were designed using Methyl Primer Express (v1.0) [58], which allows for the design of primers for bisulfite-converted DNA. Each primer pair was designed from 15 base pairs (bp) to 22 bp in length, resulting in amplicons between 250 bp and 300 bp. When CGIs were larger than 300 bp, two or more primer pairs were designed to cover as much of the CGI's sequence as possible. The size of the CGIs ranged from a minimum of 200 bp to a maximum of 2000 bp. In addition, the primers had to have at least 50% C and G contents [59].

Using BiSearch (v2.63) [60] and IDT OligoAnalyzer Tool https://eu.idtdna.com/pages/tools/oligoanalyzer (accessed on 24 November 2021) [61], primer properties and specificity were determined, and Illumina adapter overhang sequences were added to the 5' end of the sequence-specific DNA primer. The final primer sequences were checked for their properties and specificity.

### 2.3.4. Amplicon Generation and Sequencing

A modified 16S protocol from Illumina [62] was used to generate the amplicon library using two rounds of PCR to amplify the target sequences. The Eppendorf™ Mastercycler X50s 96-Well Silver Block Thermal Cycler (Eppendorf, Hamburg, Germany) was used for the reactions. A total of 25 µL of the PCR reaction volume consisted of 12.5 µL KAPA HiFi HotStart Uracil+ ReadyMix (Roche, KAPA Biosystems Ltd., Cape Town, South Africa), 1 µM primer, and 20 ng DNA. It was activated for 5 min at 95 °C, followed by 35 amplification cycles (denaturation for 30 s at 98 °C, annealing for 15 s at a temperature specific to the primer pair, and extension for 15 s at 72 °C). The final extension lasted for 1 min at 72 °C, followed by holding at 4 °C. Amplicons from the first PCR round were visualized through 2% agarose gel electrophoresis. Shorter and non-specific amplification fragments were eliminated using AMPure XP beads (Beckman Coulter, Brea, CA, USA). After purification, amplicons from each subject were combined in a tube to form an equimolar pool, and concentrations were determined using Quant-iT PicoGreen dsDNA (Thermo Scientific, Life Technologies, Waltham, MA, USA).

The second round of PCR was performed to label each subject with a uniquely identified sequence, following the 16S Metagenomic Sequencing Library Preparation protocol [62]. This allowed dual indexing during sequencing to generate uniquely tagged libraries that allowed for discrimination between the participant amplicon sequences. Pooled samples were labeled using Nextera XT Index Kit v2 Set D and Set A (Illumina, San Diego, CA, USA) with two eight-base indexes: Index 1 and Index 2. A total of 50 µL of the PCR reaction volume consisted of 25 µL KAPA HiFi HotStart Uracil+ ReadyMix (Roche, Basel, Switzerland), Nextera XT v2 primers, and 4 ng of the equimolar amplicon pool. It was activated for 45 s at 98 °C, with 10 amplification cycles (denaturation for 15 s at 98 °C, annealing for 30 s at 55 °C, and extension for 30 s at 72 °C), The final extension lasted for 1 min at 72 °C. The suitability of the length of the amplicons was further confirmed through 2% agarose gel electrophoresis.

The sequences of the forward and reverse primers for each amplicon, the sequence of the NGS adapters, the annealing temperature for each primer, and the length of each amplicon without adapters can be found in Table S2 (Supplementary Materials).

### 2.3.5. Library Preparation and NGS Sequencing

Amplicons from the second round of PCR amplification were subjected to size selection (AMPure XP paramagnetic beads, Beckman Coulter, Brea, CA, USA), followed by concentration determination. The concentration of each subject library was measured using an ultrasensitive fluorescent nucleic acid dye that allowed for the quantification of double-stranded DNA (PicoGreen dsDNA quantification reagent, Thermo Fisher, Waltham, MA, USA). The final library was prepared through equimolar pooling of the individual subject libraries to a final library with a molar concentration of 40 nM. The final library

was diluted and denatured, according to the recommendations detailed in the Illumina MiniSeq System Denature and Dilute Libraries Guide. The final library was sequenced on the Illumina MiniSeq sequencer using the MiniSeq Mid Output Kit (300 cycles) (Illumina, San Diego, CA, USA) with 150 bp paired-end reads.

*2.4. Bioinformatic and Statistical Analysis*

The FastQC tool (v0.14.1) [63] was used to check for the quality of the raw sequencing reads, while Trim galore (v0.6.7) [64] was employed to remove bases of insufficient quality (Q score below 30) and adapter sequences. Trimmed sequences were aligned to the reference genome (hg19) using Bismark (v0.21.0) [65].

The methylKit package (v3.17) [66] and the methylSig package (v3.17) [67] were used to analyze the aligned reads in the R environment (v4.0.4) [68], with the data corrected for age and multiple reads. The average DNA methylation values of the amplicons were calculated using the values of all CGIs in that amplicon (Table S3 in the Supplementary Materials).

In analyzing the nature of DNA methylation dynamics, we considered two hypotheses: that DNA methylation is a flexible phenomenon and reflects current depression symptoms, measured with the clinical scales, or that it is a cumulative change reflected with the duration of the current episode or the total duration of the disease. Therefore, to test the first hypothesis, we regarded the clinical scores (i.e., IDS-C, HAM-A, and BDI-II) as possible dependent variables, and the mean percentage of CGI methylation of each amplicon (*mBDNF_1*, *mBDNF_2*, *mBDNF_3*, *mBDNF_4*, *mBDNF_5*, *mBDNF_6*, *mBDNF_7*, *mBDNF_8*, *mBDNF_9*; *mCOMT_1*, *mCOMT_2*, *mCOMT_3*; *mSLC6A4_1*, and *mSLC6A4_2*) and age as possible independent variables. Similarly, to test the second hypothesis, we regarded the current episode duration and the total disease duration as candidate dependent variables, and the mean percentage of CGI methylation of each amplicon (*mBDNF_1*, *mBDNF_2*, *mBDNF_3*, *mBDNF_4*, *mBDNF_5*, *mBDNF_6*, *mBDNF_7*, *mBDNF_8*, *mBDNF_9*, *mCOMT_1*, *mCOMT_2*, *mCOMT_3*, *mSLC6A4_1*, and *mSLC6A4_2*) and age as possible independent variables. We included patient age as a possible independent variable, as epigenetic changes have been shown to increase with age [69,70]. To construct the actual regression model, first, we screened each predictor variable for outliers, i.e., values exceeding the first and third quartiles by 50% of the interquartile range (Figure 1). Next, the assumptions of homoscedasticity and normality were tested. Finally, we tested which of the proposed independent variables in our hypothetical models were linearly related to the dependent variables via correlation matrices. Only the predictor variables that showed a linear relation to the dependent variables were included into the regression model. In a post hoc analysis, we conducted principal component analysis (PCA) to combine the scores from the three clinical scales (i.e., IDS-C, HAM-A, and BDI-II) into a single dependent variable to test whether this would improve the predictive value of *COMT_1*. All statistical analyses were performed with R (vx64 4.1.2) [68] in RStudio (v4.1.2) [71].

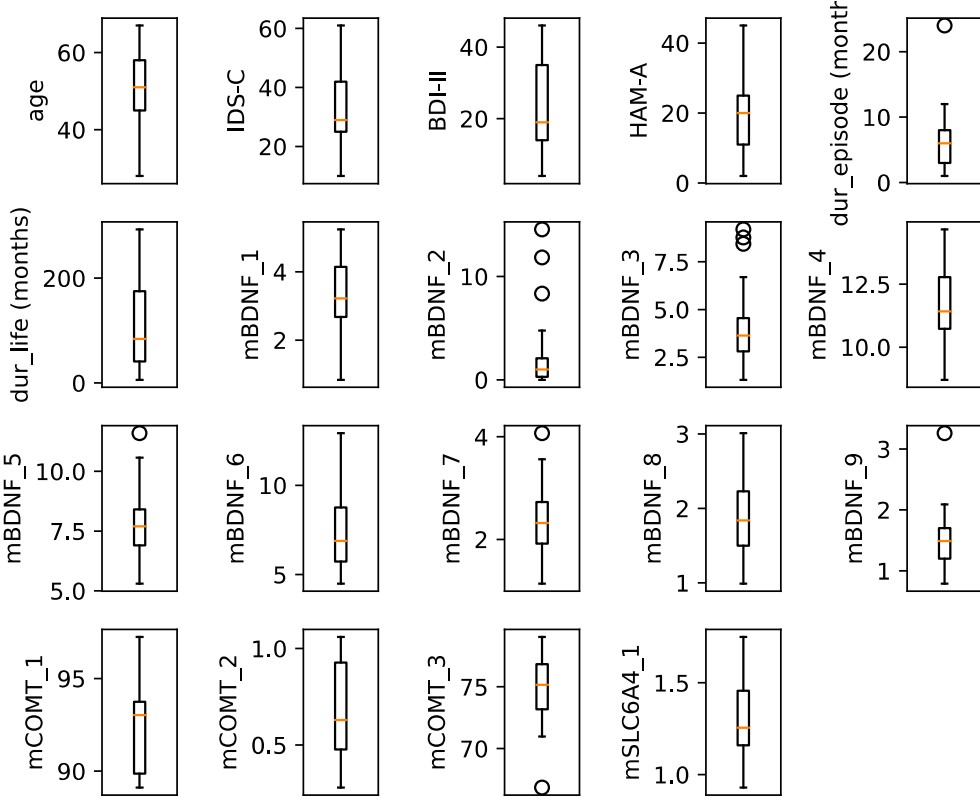

**Figure 1.** Distribution of all variables used in the models. The following dependent (clinical) variables were used: IDS-C, BDI-II, HAM-A, dur_episode (duration of the last episode in months), and dur_total (lifetime duration of the disorder). The following independent (epigenetic) variables were employed in the models: *mBDNF_1*, *mBDNF_2*, *mBDNF_3*, *mBDNF_4*, *mBDNF_5*, *mBDNF_6*, *mBDNF_7*, *mBDNF_8*, *mBDNF_9*, *mCOMT_1*, *mCOMT_2*, *mCOMT_3*, *mSLC6A4_1*, and *mSLC6A4_2*. The outliers are represented as points.

## 3. Results

### 3.1. DNA Methylation Dynamics as a Flexible Change

The preliminary model, treating DNA methylation as a dynamic and adaptive phenomenon, was constructed using linear regression models for the relationship between the clinical questionnaires BDI-II, IDS-C, and HAM-A and the mean percentage of CpG island methylation of each amplicon (*mBDNF_1*, *mBDNF_2*, *mBDNF_3*, *mBDNF_4*, *mBDNF_5*, *mBDNF_6*, *mBDNF_7*, *mBDNF_8*, *mBDNF_9*, *mCOMT_1*, *mCOMT_2*, *mCOMT_3*, *mSLC6A4_1*, and *mSLC6A4_2*), hereafter referred to as "epigenetic variables". In order to determine which epigenetic variables displayed a linear correlation with the clinical questionnaires and age, a correlation matrix was calculated. Of all the epigenetic variables, only *mCOMT_1* showed a linear relationship with all the clinical questionnaires (BDI-II, HAM-A, and IDS-C). The correlation matrices for all the epigenetic variables, clinical variables, and duration of the disorder are shown in Table 2. Second, linear regression analysis was performed to predict the clinical questionnaire scores based on the methylation level of *mCOMT_1*. The results showed a significant regression equation for BDI-II ($F(1,22) = 5.17$, $R^2 = 0.190$, $p = 0.033$), IDS-C ($F(1,22) = 5.47$, $R^2 = 0.199$, $p = 0.029$), and HAM-A ($F(1,22) = 6.59$, $R^2 = 0.231$, $p = 0.018$) (Figure 2A–C). These results suggest that *mCOMT_1* methylation can predict a significant portion of the variability observed in the BDI-II, IDS-C, and HAM-A questionnaires.

**Table 2.** Correlation matrix of the epigenetic variables (*mBDNF_1*, *mBDNF_2*, *mBDNF_3*, *mBDNF_4*, *mBDNF_5*, *mBDNF_6*, *mBDNF_7*, *mBDNF_8*, *mBDNF_9*, *mCOMT_1*, *mCOMT_2*, *mCOMT_3*, *mSLC6A4_1*, and *mSLC6A4_2*), with age as a covariate and clinical variables (BDI-II, HAM-A, and IDS-C; flexible model), as well as duration (dur_episode: duration of the current episode, and dur_total: lifetime duration of the disorder; stable model). Statistically significant *p*-values are marked in bold. The Pearson correlation coefficients (r) for *mCOMT_1* and the clinical variables display a moderate strength and negative direction. A significant linear correlation is seen for *mCOMT_1* with all clinical variables (asterisks denote uncorrected *p*-value < 0.05); n: number of participants after excluding the outliers.

| | n | Flexible Model | | | | | | Stable Model | | | |
|---|---|---|---|---|---|---|---|---|---|---|---|
| | | BDI-II | | HAM-A | | IDS-C | | Dur_Episode | | Dur_Total | |
| | | r | p | r | p | r | p | r | p | r | p |
| *BDNF_1* | 24 | 0.185 | 0.386 | 0.271 | 0.2 | 0.203 | 0.342 | 0.134 | 0.53 | −0.250 | 0.238 |
| *BDNF_2* | 21 | −0.329 | 0.145 | −0.23 | 0.316 | −0.219 | 0.341 | −0.221 | 0.377 | 0.081 | 0.748 |
| *BDNF_3* | 24 | −0.138 | 0.52 | −0.233 | 0.273 | −0.236 | 0.267 | 0.168 | 0.466 | 0.150 | 0.515 |
| *BDNF_4* | 25 | 0.13 | 0.534 | 0.117 | 0.578 | 0.049 | 0.818 | 0.201 | 0.336 | −0.192 | 0.356 |
| *BDNF_5* | 25 | 0.038 | 0.856 | 0.165 | 0.431 | 0.042 | 0.844 | 0.241 | 0.255 | −0.041 | 0.847 |
| *BDNF_6* | 24 | 0.068 | 0.753 | 0.025 | 0.909 | 0.125 | 0.559 | 0.062 | 0.774 | −0.102 | 0.634 |
| *BDNF_7* | 25 | −0.229 | 0.27 | −0.038 | 0.856 | −0.188 | 0.367 | 0.091 | 0.674 | 0.235 | 0.269 |
| *BDNF_8* | 25 | −0.067 | 0.752 | 0.146 | 0.485 | −0.221 | 0.288 | 0.002 | 0.993 | 0.221 | 0.288 |
| *BDNF_9* | 25 | −0.222 | 0.285 | −0.24 | 0.248 | −0.245 | 0.239 | 0.264 | 0.212 | −0.036 | 0.868 |
| *COMT_1* | 24 | −0.436 | **0.033 *** | −0.48 | **0.018 *** | −0.446 | **0.029 *** | −0.100 | 0.643 | 0.335 | 0.110 |
| *COMT_2* | 24 | −0.256 | 0.227 | 0.188 | 0.379 | −0.11 | 0.608 | 0.068 | 0.752 | −0.157 | 0.462 |
| *COMT_3* | 25 | −0.023 | 0.914 | −0.07 | 0.739 | −0.085 | 0.685 | −0.307 | 0.114 | 0.145 | 0.498 |
| *SLC6A4_1* | 24 | −0.144 | 0.503 | −0.064 | 0.766 | −0.256 | 0.228 | −0.004 | 0.986 | 0.119 | 0.579 |
| *SLC6A4_2* | 25 | −0.007 | 0.973 | −0.084 | 0.691 | −0.125 | 0.552 | −0.241 | 0.246 | 0.154 | 0.463 |
| Age | 25 | 0.208 | 0.318 | 0.255 | 0.218 | 0.394 | 0.051 | 0.105 | 0.614 | 0.021 | 0.919 |

### 3.2. DNA Methylation Dynamics as a Stable Change

The preliminary model, which treats DNA methylation as a stable and cumulative phenomenon, was constructed using the duration of the current episode of the disorder and the total lifetime duration of the disorder as dependent variables and epigenetic measurements as predictor variables. A correlation matrix was calculated to determine which epigenetic variables showed a linear relationship with the duration of the disorder; however, no significant relationship was found between any epigenetic variable and the duration of the current episode ($p > 0.05$) or the total lifetime duration of the disorder ($p > 0.05$) (Table 2).

### 3.3. Differential Influences of BDI-II, HAM-A, and IDS-C on the DNA Methylation Levels

In a post-hoc analysis, we aimed to investigate the contribution of the different symptom domains of the clinical picture, as measured using the clinical questionnaires (BDI-II, HAM-A, and IDS-C) to the methylation pattern of the *mCOMT_1* amplicon. First, we computed the correlation matrix between the dependent variables (BDI-II, IDS-C, and HAM-A), which revealed a significant degree of correlation between BDI-II, IDS-C, and HAM-A ($p < 0.001$), as shown in Table 3. Next, a principal component analysis (PCA) was performed to reduce the dimensionality of the data and to extract the common components. The analysis yielded a new common predictor, PC1, that explained 81.55% of the variance in our three dependent variables. All three variables contributed significantly, and in a very balanced manner, to PC1 (IDS-C: 35.37%; BDI-II: 31.52%; and HAM-A: 33.11%, see Figure 3A,B). Therefore, we used PC1 as our composite variable to reduce the number of dependent variables and to improve the predictive value of *mCOMT_1* in a linear regression model. A significant regression equation was found, $F(1,22) = 7.80$, $p = 0.011$, with an $R^2$ of 0.262, indicating that 26% of the variance in PC1 could be predicted using *mCOMT_1* (Figure 2D).

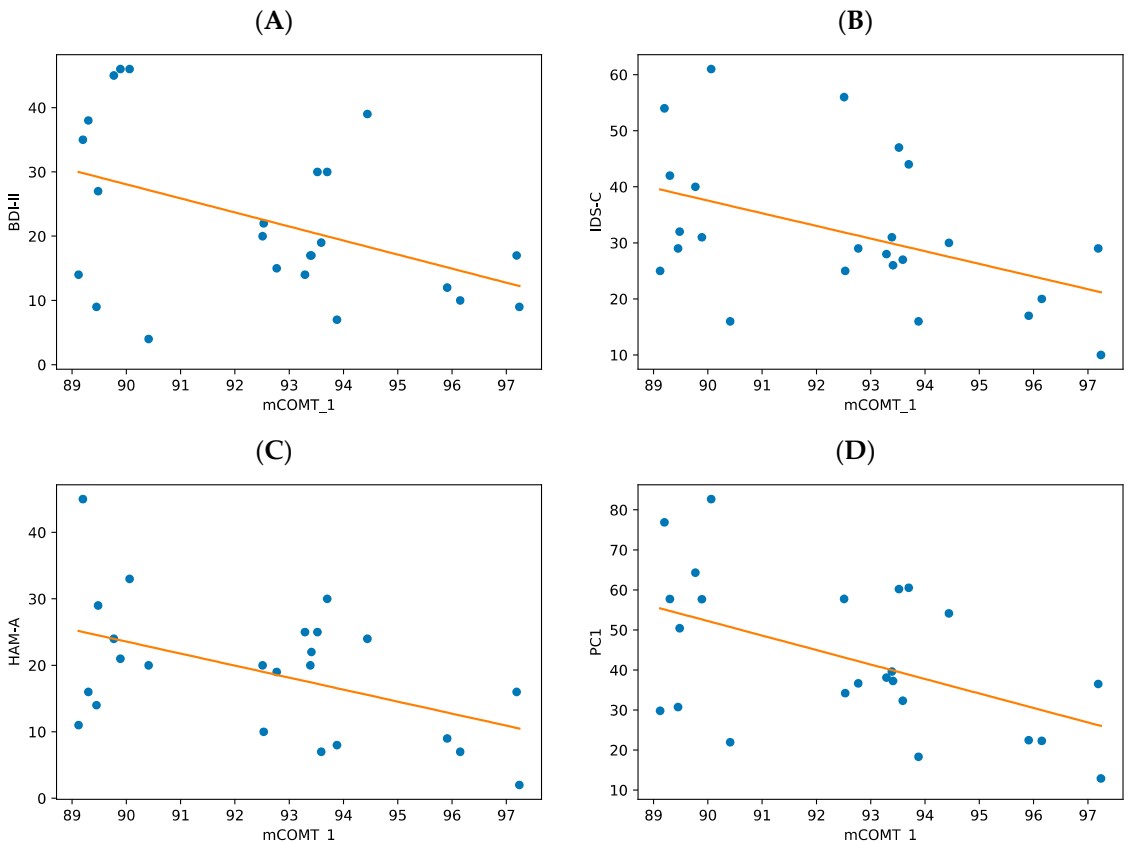

**Figure 2.** The correlation between *mCOMT_1* and the clinical variables (BDI-II, HAM-A, and IDS-C) (**A**–**C**). (**A**) BDI-II × *mCOMT_1*. A significant regression equation was found ($F(1,22) = 5.17$, $p = 0.033$), with an $R^2$ of 0.190, indicating that 19% of the variance in Beck-BDI could be predicted using *mCOMT_1*. The patient's average BDI-II score decreased by 2.19 for each unit of *mCOMT_1*. (**B**) IDS-C × *mCOMT_1*. A significant regression equation was found ($F(1,22) = 5.47$, $p = 0.029$), with an $R^2$ of 0.199, indicating that almost 20% of the variance in IDS-C could be predicted using *mCOMT_1*. The patient's average IDS-C score decreased by 2.26 for each unit of *mCOMT_1*. (**C**) HAM-A × *mCOMT1*. A significant regression equation was found ($F(1,22) = 6.59$, $p = 0.018$), with an $R^2$ of 0.231, indicating that 23% of the variance in HAM-A could be predicted using *mCOMT_1*. The patient's average HAM-A score decreased by 1.81 for each unit of *mCOMT_1*. (**D**) PC1 × *mCOMT_1*. A significant regression equation was found ($F(1,22) = 7.80$, $p = 0.011$), with an $R^2$ of 0.262, indicating that 26% of the variance in PC1 could be predicted using *mCOMT_1*. The patient's average PC1 score decreased by 0.291 for each unit of *mCOMT_1*; PC1 (first principal component).

**Table 3.** Summary of correlations between clinical variables. Shown are the values of the Pearson correlation coefficient (r) and the *p*-values (*p*).

|  | **BDI-II** | **IDS-C** | **HAM-A** |
|---|---|---|---|
| BDI-II |  |  |  |
| IDS-C | r(23) = 0.73, $p < 0.001$ |  |  |
| HAM-A | r(23) = 0.66, $p < 0.001$ | r(23) = 0.78, $p < 0.001$ |  |

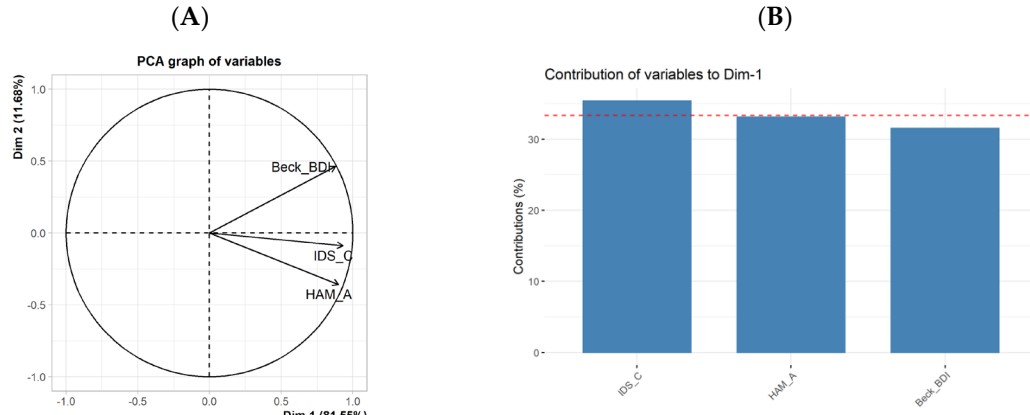

**Figure 3.** (**A**) Principal component PC1 (Dimension 1) as a summary variable of 3 clinical variables (IDS-C, BDI-II, and HAM-A). (**B**) Contributions of the clinical variables to PC1. Abbreviations: Beck-BDI: Beck Depression Inventory (BDI-II); IDS_C: Inventory of Depressive Symptomatology, Clinician version (IDS-C); and HAM_A: Hamilton Anxiety Rating Scale (HAM-A).

In summary, our results suggest that the methylation level of *mCOMT_1* is a better predictor of the composite variable PC1 than of the individual clinical variables (BDI-II, HAM-A, and IDS-C). As the components of the clinical picture (BDI-II, HAM-A, and IDS-C) are similarly reflected in the composite variable PC1, it is likely that the methylation levels are similarly related to all of them.

### 3.4. Effect of Gender on the DNA Methylation Levels

Biological sex is among the more important factors linked to stress vulnerability, making women more likely to suffer from depression and anxiety [72]. We aimed to investigate whether gender influences the methylation pattern of our candidate genes. Since only *mCOMT_1* showed a significant correlation with the clinical variables, we examined whether there is a significant difference between *mCOMT_1* and gender. We used all the data for this analysis, as no outliers were present (Figure 4). Given the small size of the groups, we first checked for equal variances using Levene's test, which showed no significant difference ($p = 0.0896$). Therefore, an independent-samples *t*-test was performed, which revealed no significant difference in the methylation levels of *mCOMT_1* by gender ($t(23) = 1.79$, $p = 0.086$). These findings indicate that gender does not have a statistically significant impact on the methylation levels of *mCOMT_1* in our dataset.

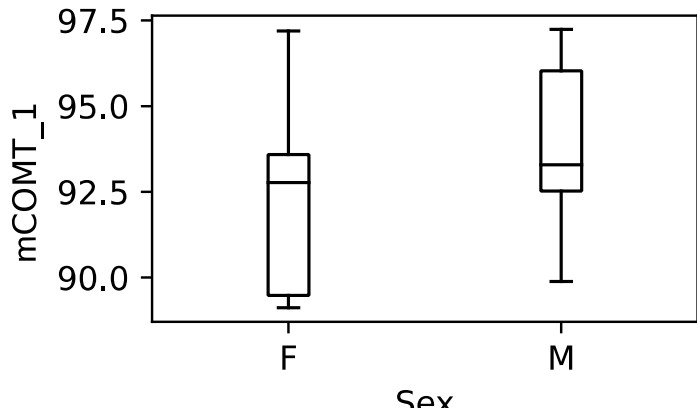

**Figure 4.** Boxplot of participants' mean methylation level of *COMT_1* amplicon separated by gender. There was no significant difference in the methylation levels ($t(23) = 1.79$, $p = 0.086$).

## 4. Discussion

Previous studies have shown that exposure to stressful situations can affect the epigenetic patterns of the epigenome [29,32]. Building on these findings, our research focused on the dynamics of DNA methylation of the candidate genes, *BDNF*, *COMT*, and *SLC6A4*, involved in the pathophysiology of stress-related disorders, i.e., pathways involved in neurotransmitter systems and neuroplasticity. This study has uncovered the following findings: (i) DNA methylation correlates with acute symptom severity, as assessed using the clinical questionnaire scores (BDI-II, HAM-A, and IDS-C), and not with the duration of the current episode or the lifetime duration of the disorder, (ii) DNA methylation contributes equally to different aspects of the clinical picture (BDI-II, HAM-A, and IDS-C), and (iii) gender has no significant effect on the methylation levels.

### 4.1. DNA Methylation as a Flexible or Stable Epigenetic Change in Anxiety and Depression

Epigenetic marks are widely recognized as useful biomarkers in somatic disease [73–76], but their applicability in mental disorders, particularly stress-related disorders, is less clear. DNA methylation has been considered a rather stable change, linking modifications at specific gene sites to social adversities that persist throughout life or even over several generations [77,78]. However, dynamic changes in DNA methylation in response to stress or as a vulnerability marker for depression have hardly been reported [79,80]. Recent research in the field of social epigenetics has shed some light on this issue. For example, studies have shown that exposure to adversity in childhood and perceived racism in adolescence can lead to stable epigenetic marks of stress-related genes. Conversely, a nurturing family environment and emotional support in childhood have been found to attenuate these epigenetic marks [81,82]. These findings suggest that DNA methylation of stress-related genes may serve as a useful biomarker for stress-related disorders and their response to environmental and lifestyle changes, potentially leading to the development of personalized treatments [79].

We examined changes in DNA methylation of the CpG islands either in the promoter or in the coding regions of our selected candidate genes, both of which may affect gene transcription [16,83] and could therefore cause observable phenotypic changes. Our results support the model that treats DNA methylation as a flexible phenomenon; although, we only found significant changes in DNA methylation for the *COMT_1* amplicon (Table 1, Figure 2; $p < 0.05$). We observed a negative correlation between *COMT_1* methylation and symptom severity, such that hypomethylation was associated with higher scores on each clinical questionnaire (BDI-II, IDS-C, and HAM-A), indicating a more severe clinical picture (Figure 2).

The relationship between the DNA methylation patterns of our candidate genes and symptom severity has been extensively investigated in several published studies. In particular, several studies have independently found a positive correlation between higher methylation levels of the *SLC6A4* promoter region and increased depressive symptoms [38,84–86]. Moreover, a longitudinal study focusing on patients with post-stroke depression found a similar association [87]. In contrast, Iga and colleagues observed a negative correlation between DNA methylation and depressive symptoms in late-life depression; although, this association did not reach statistical significance [88]. Similarly, Lam and colleagues reported similar trends, but exclusively in individuals homozygous for the short *5-HTTLPR* and *5-HTTLPR/r25531* alleles [89].

Regarding the *BDNF* gene, a higher methylation status has generally been associated with a more severe depressive episode [85,90,91]. Kang and colleagues specifically associated a higher *BDNF* promoter methylation status with a history of suicide attempts, suicidal ideation during treatment, and suicidal ideation at the last psychiatric evaluation. They also found a correlation with higher scores on the BDI scale and poor treatment outcomes for suicidal ideation [37]. On the other hand, Song and colleagues found a negative correlation between the clinical symptoms and average DNA methylation levels of the whole *BDNF* gene [92].

*COMT* hypomethylation has primarily been observed in patients with schizophrenia and bipolar disorder [93–96], and it is speculated that it may represent an epigenetic biomarker for both disorders [97] and part of the risk mechanism for decreased executive function [96]. In contrast, there are few studies on *COMT* gene methylation and stress-related disorders or depression. Na and colleagues found that *COMT* methylation was lower in patients with MDD than in healthy controls. They also observed a complex and different relationship between *COMT* methylation and prefrontal white matter connectivity in both groups [54]. Polli and colleagues found no significant association between DNA methylation of the *COMT* gene and symptoms in patients with chronic fatigue syndrome and fibromyalgia [98]. In healthy subjects, higher stress levels and lower *COMT* methylation were reported to be associated with a decreased cortical efficiency [99]. Furthermore, Wiegand and colleagues examined continuous DNA methylation changes in the *COMT* promoter region in healthy male subjects in response to a stressful cognitive task. They observed a continuous increase in methylation that correlated with higher salivary cortisol levels. The change in DNA methylation was still detectable after one week and could be reversed after transcranial direct current brain stimulation, suggesting gene-specific dynamics of DNA methylation in response to environmental stimuli [100]. Similar observations were made by Lee and colleagues, who examined DNA methylation at specific CpG sites within the *FKBP5* gene in response to corticosterone treatment. The observed DNA methylation changes were dependent on plasma corticosterone levels and returned to baseline levels following the cessation of corticosterone treatment [101].

As mentioned earlier, *COMT* is an enzyme involved in the degradation of catecholamines. Considering the effects of DNA methylation on gene expression, one could speculate that hypomethylation of our amplicon may have a regulatory function on gene expression. In this case, hypomethylation would be accompanied by an increased rate of transcription, leading to higher rates of enzymatic activity and consequently lower catecholamine levels. Interestingly, Homan et al., compared the effects of catecholamine and tryptophan depletion on brain metabolism and depressive or anxiety symptoms. Catecholamine depletion correlated with brain activity and depressive symptoms, whereas tryptophan depletion showed no significant correlation with brain activity or depressive symptoms [102]. Clinical symptoms in stress-related disorders are therefore modulated to a greater extent by the catecholaminergic signaling pathways. This may explain why the *COMT* amplicon was the only one among our group of candidate genes that showed a correlation with symptom severity. Interestingly, the *COMT_1* amplicon is located at the site of the *rs4680* (*Val158Met*) polymorphism, which has been associated with psychiatric disorders, particularly depressive disorders [103]. Similar to hypomethylation, the presence of the *Val/Val* polymorphism causes a change in the *COMT* enzyme to a more active form, causing the increased degradation of catecholamines, and is associated with the occurrence of anxiety and depression symptoms. In addition, *COMT_1* methylation may be more sensitive to acute stress, similar to what Wiegand and colleagues have observed [100], or less sensitive to other factors, such as psychopharmacological therapy.

*4.2. DNA Methylation Levels Are Similarly Associated with Symptoms of Anxiety and Depression, but Not with Gender*

As methylation levels could potentially reflect specific clinical symptoms that can vary between the acute and chronic forms of anxiety and depression, we wanted to use specific clinical questionnaires to examine whether there was a relationship between individual clinical variables and methylation. IDS-C and BDI-II address depressive symptoms, whereas HAM-A assesses anxiety. *COMT_1* methylation showed similar levels of correlation with all the clinical variables that were tested, explaining approximately 19% of the variance for BDI-II, 20% for IDS-C, and 23% for HAM-A, respectively. Given this, we wanted to test whether grouping symptoms into principal components improved or worsened the predictive value of *COMT_1* methylation. The first principal component, PC1, improved the predictive value of *COMT_1* methylation, as it explained up to 26% of the variance.

Thus, these results suggest that the anxiety and depression symptom domains are less likely to show independent relations with *COMT* gene methylation. Thus, our second hypothesis, that clinical variables have an independent and differential relation with DNA methylation, was not confirmed. These results add to the literature supporting the existence of common mechanisms in depression and anxiety.

In addition, we investigated whether *COMT* values were different for the two genders assessed, as this has been shown to be an important factor in vulnerability to mental disorders, with women being more prone to depression and anxiety disorders [104,105]. Estrogen has been shown to trigger epigenetic changes by affecting de novo methylation through the activation of the DNMT enzymes and demethylation through the involvement of the TET enzymes [106]. Our data showed no significant difference in *COMT_1* methylation when we compared the female and male subjects in this sample (Figure 4).

*4.3. Limitations*

Several aspects must be considered when interpreting our results. First, our sample was small, resulting in a low statistical power. Therefore, it is possible that additional associations between the clinical variables and epigenetic markers might be found in future studies with larger samples. In addition, the sample used was quite heterogeneous and unevenly represented in terms of age and gender, which we tried to take into account in our analysis. Second, due to the nature of our research problem, which was to compare acute and chronic anxiety and depressive disorders, we included patients who may have substantially differed in terms of their psychotropic treatment. Given the small and heterogeneous sample size, it was not possible to test the effects of the type of therapy on the methylation levels. Several genes, including *BDNF* and *SLC6A4*, show interactions between DNA methylation and the use of psychotropic medications [107], such as antidepressants [108,109]. The results of our study confirmed an association between DNA methylation and the severity of clinical symptoms rather than the duration of anxiety and depression. However, it is crucial to acknowledge that this conclusion only applies to the specific amplicon analyzed in our study. While it is possible that there are significant changes in DNA methylation throughout the gene sequence, much of the DNA methylation was generally located in the CpG islands in the promoter region that we studied.

## 5. Conclusions

Epigenetic mechanisms represent a sophisticated apparatus that allows for stable and dynamic changes to coexist in a gene-specific manner. The precise mechanisms controlling these gene-specific changes remain elusive. By understanding the common mechanisms underlying depression and anxiety, we can develop more individualized and effective interventions that address the complex nature of comorbid symptoms. Further research is needed to explore the broader epigenetic landscape and its relationship to phenotypic changes in response to stress over time.

**Supplementary Materials:** The following supporting information can be downloaded at: https://www.mdpi.com/article/10.3390/cimb45090461/s1, Table S1: The clinical characteristics of the participants; Table S2: Primer sequences (highlighted), NGS adapter (bold), annealing temperature, and amplicon length without adapters; N designates any base; Table S3: Average DNA methylation values of amplicons; n/a: information not available.

**Author Contributions:** Conceptualization, E.V., S.P., B.N.Š., A.V.P. and J.B.; methodology, A.V.P., M.A., K.K., I.P. and J.B.; formal analysis, M.A., K.K., I.P. and A.V.P.; investigation, E.V., S.P., B.N.Š. and J.B.; resources, A.V.P. and J.B.; data curation, E.V., S.P., I.P. and J.B.; writing—original draft preparation, E.V., S.P., A.O., B.N.Š., A.V.P. and J.B.; writing—review and editing, E.V., S.P., A.O., B.N.Š., K.K., A.V.P. and J.B.; visualization, I.P.; supervision, B.N.Š., A.V.P. and J.B.; project administration, E.V. and S.P.; funding acquisition, A.V.P. and J.B. All authors have read and agreed to the published version of the manuscript.

**Funding:** This research was funded by Javna Agencija za Raziskovalno Dejavnost RS (grants P5-0110, P1-0390, J3-1763).

**Institutional Review Board Statement:** The study was conducted in accordance with the Declaration of Helsinki and approved by the Medical Ethics Committee of the Republic of Slovenia (protocol code 0120-692/2015-2, date of approval 16 December 2015).

**Informed Consent Statement:** Informed consent was obtained from all subjects involved in the study.

**Data Availability Statement:** The data presented in this study are available on reasonable request from the corresponding author. The data are not publicly available due to privacy and ethical restriction.

**Acknowledgments:** The authors want to thank the participants, without whom this study would not have been possible.

**Conflicts of Interest:** The authors declare no conflict of interest. The funders had no role in the design of the study; in the collection, analyses, or interpretation of data; in the writing of the manuscript; or in the decision to publish the results.

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
