# Peer review of "DNA Methylation Patterns in Relation to Acute Severity and Duration of Anxiety and Depression"

_cimb, doi:10.3390/cimb45090461_

Round 1

Reviewer 1 Report

Overall, the manuscript is well written. The authors are advised to double check the robustness and statistical analysis. Method and discussion need to be more elaborative.

Reviewer 2 Report

This is a good study that correlates methylation and cognitive research. However, the authors should have performed some modifications prior to acceptance.

1)      The abstract is written descriptively and does not have numerical results. I strongly suggest adding numerical results to enrich it.

2)      Why the authors chose these three genes: BDNF, COMT, and SLC6A4?

3)      I suggest adding a graphical abstract at the end of the introduction part.

4)      Is there any overlap between Table 2 and Figure 2? In this case, they should move Table 2 into the supplementary materials.

5)      Hoe the authors prove DNA methylation levels are influenced by symptoms of anxiety and depression? For this purpose, depression level must reproducibility be quantified. (line 488)

6)      Doe DNA methylation happens in the promoter region or entire the gene?

7)      The author claimed there is a negative correlation between COMT methylation and symptom severity. (lines 430-432) Then, they mentioned hypomethylation is associated with worsening clinical symptoms.  These two outcomes are not consistent. Hypomethylation means less methylation and if the negative correlation was true, it should correlate with better clinical status.

Reviewer 3 Report

DNA methylation of stress-related genes correlates, although in a limited number of patients, with severity of anxiety and depression, which may serve as a biomarker for diagnosis and personalized therapy of  these disorders.

Significant data for the practice, that deserve to be published and replicated in larger cohort.

Round 2

Reviewer 2 Report

The Authors answered my questions and the quality of the work has been improved. I would recommend publishing.